# Ultrasound-guided dynamic needle tip positioning technique for radial artery cannulation in elderly patients: A prospective randomized controlled study

**Soo Yeon Kim, Kyu Nam Kim** ⓘ *, **Mi Ae Jeong, Bong Soo Lee, Hyun Jin Lim**

Department of Anesthesiology and Pain Medicine, Hanyang University Hospital, College of Medicine, Seoul, Republic of Korea

* vesicle100@naver.com

## Abstract

### Background

Radial artery cannulation, which is a useful procedure for anesthetic management, is often challenging in elderly patients. Recently, the dynamic needle tip positioning (DNTP) technique was introduced to facilitate ultrasound-guided vascular catheterization. Therefore, we performed this prospective, parallel group, randomized, controlled trial to compare the ultrasound-guided DNTP technique with the palpation method in elderly patients.

### Methods

For this study, 256 patients aged 65 years or older were randomly allocated to the ultrasound-guided DNTP technique group (DNTP group) or the palpation method group (palpation group). The primary outcome was first-attempt success rate. The secondary outcomes were overall success rate, numbers of attempts and redirections, cannulation time, and incidence of complications.

### Results

The first-attempt success rate (85.9% vs. 72.3%; relative risk [RR], 1.47; 95% CI 1.25–1.72; $P<0.001$) and the overall success rate (99.2% vs. 93.0%; RR, 1.07; 95% CI 1.02–1.12; $P = 0.01$) were significantly higher in the DNTP group compared to the palpation group. The numbers of attempts [1 (1,1) vs. 1 (1,3), $P<0.001$] and redirections [0 (0,1) vs. 2 (0,4), $P<0.001$] were significantly lower in the DNTP group. The cannulation time for successful attempts was 42 (32,55) seconds in the DNTP group and 53 (36,78) seconds in the palpation group ($P<0.001$). The incidence of hematoma was significantly lower in the DNTP group (7% vs. 24.2%; RR, 0.29; 95% CI, 0.14–0.59; $P<0.001$).

**Data Availability Statement:** The datasets used and analysed during the current study are available in S5 Original data set. (Supporting information).

**Funding:** The authors received no specific funding for this work.

**Competing interests:** The authors have declared that no competing interests exist.

## Conclusions

Ultrasound-guided radial artery cannulation with the DNTP technique improved the efficiency of radial artery cannulation in elderly patients by increasing the success rate while minimizing complications.

## Introduction

Arterial cannulation is a procedure often performed for repetitive blood tests and real-time monitoring of patient blood pressure during surgery [1]. The arterial line can be placed at various locations, but the radial artery is the most commonly used blood vessel due to its easy accessibility and presence of dual supply to the hands through the ulnar artery [2]. As elderly patients show decreased response to beta-receptor stimulation and increased systemic vascular resistance and sympathetic nervous system activity, they often show non-stable blood pressure and heart rate during anesthesia [3]. However, arterial cannulation is often difficult in the hard-to-catheterize radial artery in elderly patients because they often have age-related arterial wall changes and tortuous arteries due to various underlying diseases [4–6].

Ultrasound can be applied with a short-axis view of the targeted artery using an out-of-plane approach (SAX-OOP approach) for arterial cannulation. Although the SAX-OOP approach offers a better view of surrounding structures during a targeted vessel approach, the posterior wall puncture rate is significantly higher [7, 8]. This is because the needle is visualized only as a dot in the SAX-OOP approach, and the plane of the ultrasound may pass through the shaft rather than the tip of the needle [8].

To compensate for this drawback of the SAX-OOP approach, the dynamic needle tip positioning (DNTP) technique was introduced [9]. With DNTP, the probe moves along the arteries in small increments, and the needle is advanced in the same direction. By applying this technique to the SAX-OOP approach, practitioners can trace the needle tip more accurately. Although several studies have compared the use of ultrasound with the conventional palpation method for radial artery cannulation [10–18], none of them investigated the efficacy and safety of the ultrasound-guided DNTP technique in elderly patients. We hypothesized that the DNTP technique would have a higher success rate and lower incidence of complications compared with the conventional palpation method. Therefore, we performed this prospective, parallel group, randomized, controlled trial to evaluate the efficacy and safety of the ultrasound-guided DNTP technique.

## Materials and methods

This study was performed by CONSORT guidelines. After approval by the Hanyang University Hospital Institutional Review Board (approval number: HYUH 2018-10-024-001), this study was prospectively registered with the Clinical Research Information Service (website: https://cris.nih.go.kr/; registration number: KCT0003507). The purpose and procedures of the study were explained to eligible patients, and written informed consent was obtained. Recruitment of participants began on March 6, 2019 and follow-up ended on July 29, 2019.

### Selection of participants

Patients 65 or older who were undergoing general anesthesia for surgeries that required arterial catheterization and were American Society of Anesthesiologist (ASA) classification I, II, or

III were included in this study. Patients were excluded if they were hemodynamically unstable (systolic blood pressure 60 or less), or if they had skin abnormalities such as inflammation or hematoma at the cannulation site. Patients were also excluded if they showed abnormal results on the modified Allen test or had history of hand or wrist surgery.

## Randomization and allocation concealment

Patients enrolled in the study were allocated to either the ultrasound-guided DNTP technique group (DNTP group) or the palpation method group (palpation group) with a 1:1 ratio. Randomization was performed by an independent person using a computer-generated random number list. From a total of 256 patients, 128 were allocated to each group (Fig 1). The allocation results were sealed in envelopes that were opened just before artery cannulation.

## Blinding

It was not possible to blind cannulation practitioners to method used. However, enrolled participants were blinded, and a separate observer who was blinded to patient group measured the diameter and depth of the radial artery and recorded the outcomes. A barrier was placed between the practitioner and the outcome observer.

## Interventions

After patients entered the operating room, standard monitoring of pulse oximetry, non-invasive blood pressure measurements, and electrocardiography was applied. The cannulation practitioner selected the right or left arm for radial artery cannulation depending on surgery site, location of blood pressure cuff, and his/her preference. The modified Allen test was performed and a positive test result was considered with adequate collateral blood flow from the ulnar artery. The wrist was placed on a soft roll for mild dorsiflexion. At the level of the radial styloid process, the diameter and depth of the radial artery were measured using ultrasound equipment, and the mean of two consecutive measurements was used for analysis. The cannulation practitioner was blinded to the ultrasound images and measurement values. In this study, we used a Sonosite M-Turbo (Bothell, WA, USA) ultrasound machine with a linear transducer probe (HFL 38X/13-6 MHz).

Anesthesia was induced with 1% lidocaine, propofol, and rocuronium based on our routine protocol. When vital signs were stable, radial artery cannulation was performed using either the ultrasound-guided DNTP technique or the palpation method under aseptic conditions before endotracheal intubation.

**Ultrasound-guided DNTP technique.** The cannulation practitioner identified the radial artery with an ultrasound probe in the SAX-OOP approach based on the motion of the radial artery or color Doppler as needed. After the radial artery was centered in the ultrasound display, a 22-gauge angiocatheter (BD Angiocath Plus, Becton Dickinson Medical Pte Ltd., Singapore) was inserted through the skin under the midline of the probe until the needle tip appeared as a hyperechoic dot on the display. Then the practitioner moved the probe slightly further without moving the needle. When the hyperechoic dot disappeared as the needle tip exited the ultrasound plane, the needle was advanced toward the radial artery with the probe held in place. Once the needle punctured the radial artery, the procedure was repeated, leaving the needle tip in the center of the radial artery. After confirmation that the needle tip remained in the radial artery, the outer catheter was pushed to the end, and the core needle was removed. The process is described with ultrasound images in Fig 2.

**Palpation method.** The cannulation practitioner palpated the radial arterial pulse with one hand. After the arterial pulse was confirmed, a 22-gauge angiocatheter was inserted at an

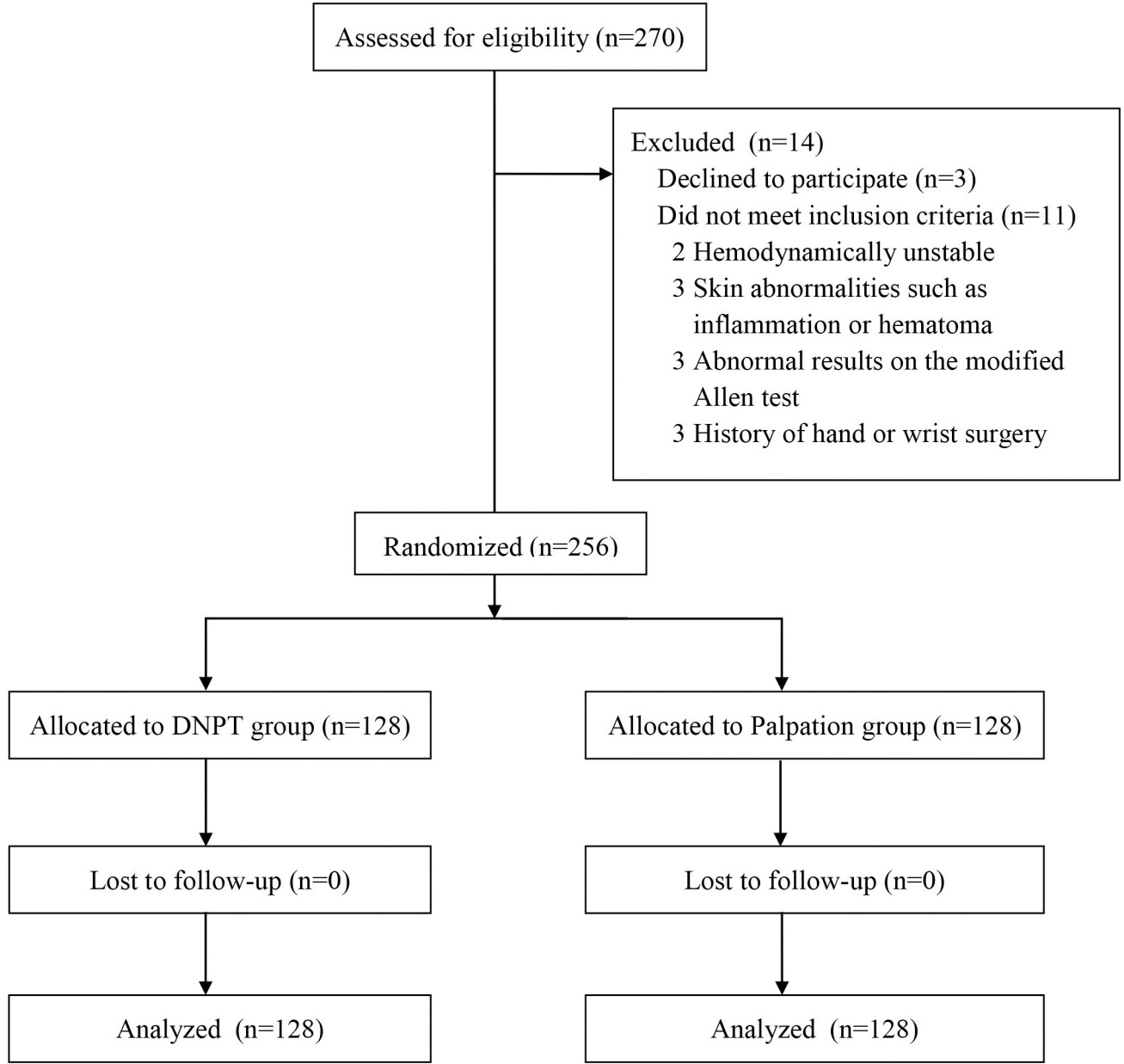

**Fig 1. Flow diagram of patient recruitment and exclusion criteria for the study.** DNTP, dynamic needle tip positioning technique.

angle of 30 to 45 degrees within the skin. When the radial artery was punctured and blood appeared in the catheter hub, the practitioner slightly reduced the angle and advanced the needle a few millimeters. Then, the outer catheter was pushed to the end, and the core needle was removed.

In this study, four residents in their second year of the four-year training were chosen as cannulation practitioners. They all practiced at least 40 radial arterial cannulations using the ultrasound-guided DNTP and palpation methods before the first patient enrollment. The number of prerequisite cases for each procedure was based on previous literature, which identified learning curves for radial artery cannulation [19, 20] and ultrasound needle visualization [21], as well as recommendations for ultrasound training [22].

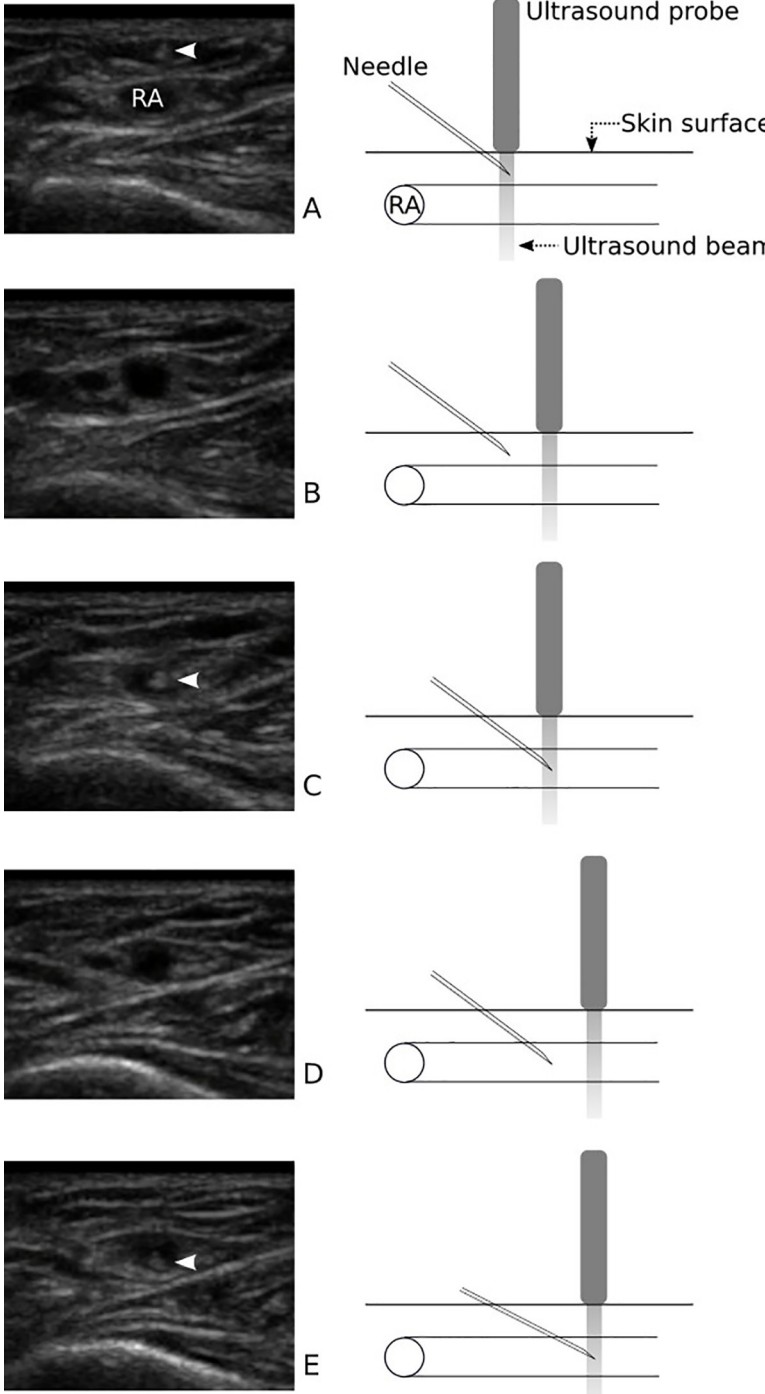

**Fig 2. Ultrasound-guided DNTP technique.** The radial artery (RA) was identified in an out-of-plane approach with an ultrasound probe. (A) The needle was inserted through the skin, and the needle tip (*arrowhead*) appeared as a hyperechoic dot on the display. (B) The probe was moved slightly further without moving the needle, and the hyperechoic dot disappeared as the needle tip exited the ultrasound plane. (C) The needle was advanced toward the radial artery with the probe held in place, and the needle punctured the radial artery. (D) The probe was moved proximally again, and the needle tip disappeared from the display. (E) The needle was advanced, and it was confirmed that the needle tip remained in the radial artery. *Arrowheads* on the ultrasound images indicate the needle tip of the angiocatheter. DNTP, dynamic needle tip positioning.

## Measured outcomes

The primary outcome was first-attempt success rate. The secondary outcomes were overall success rate, numbers of attempts and redirections, cannulation time, and incidence of complications (hematoma, thrombosis, spasm, and ischemia).

Success was confirmed when an arterial waveform was observed on the vital signs monitor. An attempt was defined as needle penetration of the skin. After the fifth attempt, the procedure was considered a failure, and the practitioner was free to select either method. Redirection was defined as pulling the needle back slightly and changing the direction without exiting the skin. The timer was started when the probe was attached to the skin in the DNTP group and when the practitioner started palpation of the radial artery in the palpation group. The timer was stopped when an arterial waveform appeared on the vital signs monitor. The cannulation time for the last attempt was recorded. Hematoma was defined as a localized swelling around the cannulation site [17]. Thrombosis was confirmed when a clot in the radial artery was detected by ultrasound [17]. Spasm was identified by the operator as any significant resistance [23]. Ischemia was defined as pallor after the procedure [24].

## Sample size calculation

The sample size was calculated by referring to a randomized controlled trial [10] comparing ultrasound-guided DNTP and the palpation method for radial artery cannulation. The first-attempt success rates were 83% and 48%, respectively. Therefore, we expected that the first-attempt success rate would be 50% using the palpation method in our study. We assumed that it would be meaningful when the success rate increased by 20% with the ultrasound-guided DNTP technique. To detect a difference in the success rate using a $\chi^2$ test with a power of 90% and a significance level ($\alpha$) of 0.05, a total of 253 subjects was required. Therefore, 256 subjects (128 in each group) were enrolled to account for a 1% dropout rate (PASS 14 Power Analysis and Sample Size Software, 2015. NCSS, LLC. Kaysville, UT, USA).

## Statistical analysis

Categorical data were expressed as number of patients (percentage) and compared using Pearson's $\chi^2$ test or Fisher's exact test. Continuous data were compared using Student's t-test or the Mann-Whitney U-test and expressed as mean (standard deviation) or median (first, third quartiles), respectively. A normality test for continuous data was performed using the Shapiro-Wilk test. Multivariable logistic regression analysis was used to identify factors associated with first-attempt success rate. $P$ values $< 0.05$ were considered statistically significant. Statistical analysis was performed with SPSS software (version 25; SPSS, Chicago, IL, USA).

## Results

Among 270 patients who were assessed for eligibility, 14 were excluded. Three patients declined to participate, and 11 patients did not meet inclusion criteria. Consequently, a total of 256 patients was enrolled in this study, and 128 were randomly assigned to each group (Fig 1). Patient demographic data are summarized in Table 1. The mean diameter of the radial artery was 2.3 (0.4) mm in the DNTP group and 2.3 (0.5) mm in the palpation group ($P = 0.70$), and the mean depth of the radial artery was 2.6 (2.0, 3.6) mm and 2.6 (2.0, 3.4) mm, respectively ($P = 0.63$).

The first-attempt success rate was significantly higher in the DNTP group (85.9%) compared to the palpation group (72.3%; relative risk [RR], 1.47; 95% confidence interval [CI] 1.25–1.72; $P < 0.001$; Table 2). The overall success rate was also higher in the DNTP group

**Table 1. Patient characteristics.**

|  | DNTP Group (n = 128) | Palpation Group (n = 128) | *P* value |
|---|---|---|---|
| Age, years | 72 (68, 78) | 74 (69, 78) | 0.32 |
| Sex, female | 85 (66.4) | 82 (64.1) | 0.69 |
| BMI, kg m$^{-2}$ | 24.0 ± 4.0 | 23.8 ± 3.5 | 0.75 |
| ASA (I/II/III) | 8/89/31 (6.3/69.5/24.2) | 8/95/25 (6.3/74.2/19.5) | 0.66 |
| Hypertension, n | 86 (67.2) | 92 (71.9) | 0.42 |
| Diabetes mellitus, n | 37 (28.9) | 32 (25.0) | 0.48 |
| Hypercholesterolemia, n | 35 (27.3) | 37 (28.9) | 0.78 |
| Peripheral vascular disease, n | 28 (21.9) | 18 (14.1) | 0.10 |
| History of smoking, n | 13 (10.2) | 8 (6.3) | 0.26 |
| Diameter of the RA, mm | 2.3 ± 0.4 | 2.3 ± 0.5 | 0.70 |
| Depth of the RA, mm | 2.6 (2.0, 3.6) | 2.6 (2.0, 3.4) | 0.63 |
| SBPstart, mmHg | 120.0 ± 22.3 | 119.7 ± 21.4 | 0.91 |
| HRstart, bpm | 77 (65.0, 84.8) | 75.5 (65.3, 88.8) | 0.31 |

Values are presented as number (percentage, %), mean ± standard deviation or median (first, third quartiles).

DNTP, dynamic needle tip positioning; BMI, body mass index; ASA, American Society of Anesthesiologist physical status classification; n, number; RA, radial artery;

SBPstart, systolic blood pressure at the start of the procedure; HRstart, heart rate at the start of the procedure; bpm, beats per minute.

(99.2% vs. 93.0%; RR, 1.07; 95% CI 1.02–1.12; *P* = 0.01). The number of attempts was 1 (1, 1) in the DNTP group and 1 (1, 3) in the palpation group (*P* < 0.001). The number of redirections for successful attempts was 0 (0, 1) in the DNTP group and 2 (0, 4) in the palpation group (*P* < 0.001, Table 2). The cannulation time for successful attempts was 42 (32, 55) seconds and 53 (36, 78) seconds (*P* < 0.001), respectively.

In terms of complications, the incidence of hematoma was significantly lower in the DNTP group (7%) compared to the palpation group (24.2%; RR, 0.29; 95% CI, 0.14–0.59; *P* < 0.001; Table 3). Thrombosis occurred in one patient in the palpation group and resolved without further complications after one week. No patients had spasm or ischemia. The number of attempts and the incidence of hematoma were analyzed in detail (Table 4). When cannulation was successful on the first attempt, hematoma was not observed. Of 71 cases that needed two or more attempts for radial artery cannulation, hematoma developed in 40 (56.3%).

**Table 2. Comparison of outcome data of all attempted cases and successful cases.**

|  | Variables | DNTP Group | Palpation Group | *P* value | Relative Risk (95% CI) |
|---|---|---|---|---|---|
| All attempted cases | Number of cases | 128 | 128 |  |  |
| . | First-attempt success, n | 110 (85.9) | 75 (72.3) | < 0.001 | 1.47 (1.25 to 1.72) |
|  | Overall success, n | 127 (99.2) | 119 (93.0) | 0.01 | 1.07 (1.02 to 1.12) |
|  | Number of attempts | 1 (1, 1) | 1 (1, 3) | < 0.001 |  |
| Successful cases | Number of cases | 127 | 119 |  |  |
|  | Number of redirections | 0 (0, 1) | 2 (0, 4) | < 0.001 |  |
|  | Cannulation time, second | 42 (32, 55) | 53 (36, 78) | < 0.001 |  |
|  | SBPend, mmHg | 104.4 ± 20.7 | 106.3 ± 19.8 | 0.47 |  |
|  | HRend, bpm | 74 (62, 82) | 73 (62, 91) | 0.23 |  |

Values are presented as number (percentage, %), mean ± standard deviation or median (first, third quartiles).

DNTP, dynamic needle tip positioning. SBPend, systolic blood pressure at the end of the procedure; HRend, heart rate at the end of the procedure; bpm, beats per minute.

**Table 3. Complications after radial artery cannulation.**

|  | DNTP Group (n = 128) | Palpation Group (n = 128) | *P* value | Relative Risk (95% CI) |
|---|---|---|---|---|
| Hematoma, n | 9 (7.0) | 31 (24.2) | < 0.001 | 0.29 (0.14 to 0.59) |
| Thrombosis, n | 0 (0.0) | 1 (0.8) | 1.000 | 0.33 (0.01 to 8.11) |
| Spasm, n | 0 (0.0) | 0 (0.0) |  |  |
| Ischemia, n | 0 (0.0) | 0 (0.0) |  |  |

Values are presented as number (percentage, %).

DNTP, dynamic needle tip positioning; n, number.

We tried to identify other factors that might have affected first-attempt success for radial artery cannulation. After multivariable logistic regression analysis, use of ultrasound (odds ratio [OR], 4.33; 95% CI, 2.34–8.00; $P < 0.001$) and diameter of the radial artery (OR, 2.15; 95% CI, 1.08–4.30; $P = 0.03$) were associated with first-attempt success (S1 Table).

## Discussion

In this study, the first attempt success rate was significantly higher in the DNTP group compared to the palpation group. In addition, use of ultrasound provided a better overall success rate and improved the numbers of attempts and redirections, cannulation time, and complications.

Several studies have compared an ultrasound-guided method with palpation for radial artery cannulation in adult patients [10–16] and children [17, 18]. Berk et al. [25] compared the SAX-OOP and long axis views with in-plane (LAX-IP) approaches for ultrasound-guided radial artery cannulation and found that the LAX-IP approach increased the first attempt success rate (76%) compared to the SAX-OOP approach (51%) in adult patients. However, in infants and children [26], the first-attempt rate did not differ significantly between the two approaches (58.0% for the SAX-OOP approach and 54.9% for the LAX-IP approach).

In addition to these two approaches, the DNTP technique was introduced to assist ultrasound-guided vascular catheterization. Clemmesen et al. [9] first showed that the SAX-OOP approach with the DNTP technique was superior to the LAX-IP approach for peripheral vascular access in a phantom study. With the DNTP technique, the success rate was higher (97% vs. 81%) and the distance from the center of the vessel to the final needle tip position was

**Table 4. Exploratory analysis of number of attempts and incidence of hematoma.**

| Number of attempts | DNTP group (n = 128) | Hematoma, n | Palpation group (n = 128) | Hematoma, n | Total (n = 256) | Hematoma, n |
|---|---|---|---|---|---|---|
| 1 | 110 (85.9) | 0 | 75 (58.6) | 0 | 185 (72.3) | 0 |
| 2 | 11 (8.6) | 3 | 17 (13.3) | 6 | 28 (10.9) | 9 |
| 3 | 6 (4.7) | 5 | 18 (14.1) | 12 | 24 (9.4) | 17 |
| 4 | 0 (0.0) | 0 | 10 (7.8) | 8 | 10 (3.9) | 8 |
| 5 | 1 (0.8) | 1 | 6 (4.7) | 4 | 7 (2.7) | 5 |
| 6 | 0 (0.0) | 0 | 2 (1.6) | 1 | 2 (0.8) | 1 |
| N ≥ 2* | 18 (14.1) | 9 (50) | 53 (41.4) | 31 (58.5) | 71 (27.7) | 40 (56.3) |

Values are presented as number (percentage, %).

DNTP, dynamic needle tip positioning; n, number.

*For cases that required two or more attempts, 40 (56.3%) out of 71 cases developed a hematoma after radial artery cannulation, including 9 (50%) out of 18 cases in the DNTP group and 31 (58.5%) out of 53 cases in the palpation group.

shorter compared to the LAX-IP approach. Therefore, we selected the SAX-OOP approach with the DNTP technique for ultrasound-guided radial artery cannulation.

This study revealed that the first-attempt success rate was higher in the DNTP group (85.9%) than the palpation group (72.3%). This result was consistent with previous studies [10, 15, 17, 27]. The previously reported first-attempt success rates with the DNTP technique in adult patients were 83%[10] and 95% [15]. Grandpierre et al. [28] showed that use of ultrasound improved the first-attempt success rate for radial artery puncture in patients with difficult-to-obtain radial arterial blood gas analysis, as defined by non-palpable radial arteries or two previous puncture failures. The use of ultrasound makes it more feasible to identify the radial artery in cases requiring multiple attempts. The process of radial artery cannulation requires successful radial artery puncture in addition to advancement of the catheter. In many cases, failure is due to unsuccessful advancement of the catheter even if puncture of the artery is achieved. Application of the DNTP technique to the SAX-OOP approach can help the needle tip be accurately located in the radial artery by subsequent confirmation after puncture of the radial artery. This was confirmed by Takeshita et al. [27], who showed that addition of the DNTP technique to the SAX-OOP approach significantly reduced posterior wall punctures in small children with a radial artery. Together, these results demonstrate that use of ultrasound with the DNTP technique can aid both puncture of the radial artery and advancement of the catheter.

The palpation method had a higher first-attempt success rate (72.3%) than expected based on a previous study (48%) [10]. This was probably due to the difference in number of radial arterial cannulations performed by the practitioners before the start of the study. The practitioners in the previous study [10] placed at least 10 radial arterial cannulations using the DNTP technique and the palpation method prior to participation. The cannulation practitioners in our study performed arterial cannulation in at least 40 cases with either method to become familiar with the techniques. The minimum number of cases required was determined by previously reported learning curves and recommendations for ensuring patient safety [19–22].

In previous meta-analyses [29, 30], the incidence of hematoma did not differ between the ultrasound-guided method and the palpation method for radial artery cannulation. To our knowledge, no studies have identified the incidence of hematoma with the ultrasound-guided DNTP technique in adult patients. In this study, the incidence of hematoma was significantly lower in the DNTP group (7% vs. 24.2%). Notably, in both groups, hematoma developed in more than half of the cases requiring more than one attempt (40 of 71 cases; 56.3%, Table 4). Hematoma at the puncture site can interfere with subsequent attempts and is associated with an increased incidence of occlusion [1, 2]. Moreover, catheterization may become more difficult after a failed attempt due to arterial spasm [25, 31]. Therefore, it may be more important than we think to successfully carry out radial artery cannulation on the first attempt to prevent multiple attempts and subsequent tissue damage.

The factors that might be associated with first-attempt success were evaluated by logistic regression analysis. The use of ultrasound was the most powerful factor in increasing the success rate on the first attempt, and the diameter of the radial artery also affected the first-attempt success rate. However, caution is needed in interpretation because extreme cases such as severe hemodynamic instability were excluded from our study. More research is needed to discern the risk factors of arterial cannulation.

## Limitations of the study

There are several limitations in this study. First, the cannulation practitioners could not be blinded to the cannulation method used. This might induce potential bias that could arise

from a participant's expectations. However, separate observers were blinded to the cannulation method to minimize bias. Second, the practitioners who performed radial artery cannulation were limited to residents at the same grade and likely do not represent novices or experts. However, Kiberenge et al. [10] demonstrated no significant difference between anesthesia residents, fellows, and faculty in the first-attempt and overall success rates for radial artery cannulation with the ultrasound-guided DNTP technique or the palpation method. Therefore, we hypothesize that the results of this study can be extended to other practitioners with different careers. Third, we defined patients who were 65 or older as elderly following many other medical studies [32–34]. If we set the elderly population to be 75 years or older, age-related arterial wall changes and tortuous arteries would have been more prevalent in the study population. Lastly, although spasm was not reported by the operators, we might have missed some radial artery spasms because the definition of spasm used in this study was based on subjective criteria.

Despite these limitations, this prospective, randomized controlled trial revealed for the first time the efficacy and safety of the ultrasound-guided DNTP technique in elderly patients.

## Conclusions

Ultrasound-guided radial artery cannulation with the DNTP technique significantly improved the first-attempt and overall success rates and reduced the numbers of attempts, redirections, cannulation time, and complications in elderly patients compared to the conventional palpation method. We expect that use of ultrasound with DNTP will increase the efficiency of radial artery cannulation and minimize tissue damage and complications by reducing the number of needle passes in elderly patients.

## Supporting information

**S1 Checklist. Consort checklist.**
(DOC)

**S1 Table. Logistic regression analysis of factors related to first-attempt success using the backward likelihood ratio method.**
(DOCX)

**S1 Text. Ethical committee approval.**
(PDF)

**S1 File. Original protocol.**
(DOC)

**S2 File. Original data set.**
(XLSX)

## Author Contributions

**Conceptualization:** Soo Yeon Kim, Kyu Nam Kim.

**Data curation:** Soo Yeon Kim, Kyu Nam Kim, Bong Soo Lee, Hyun Jin Lim.

**Formal analysis:** Soo Yeon Kim, Kyu Nam Kim, Mi Ae Jeong.

**Investigation:** Soo Yeon Kim, Kyu Nam Kim, Bong Soo Lee, Hyun Jin Lim.

**Methodology:** Soo Yeon Kim, Kyu Nam Kim, Mi Ae Jeong, Bong Soo Lee, Hyun Jin Lim.

**Project administration:** Soo Yeon Kim, Kyu Nam Kim.

**Resources:** Soo Yeon Kim, Kyu Nam Kim.

**Supervision:** Soo Yeon Kim, Kyu Nam Kim, Mi Ae Jeong.

**Visualization:** Soo Yeon Kim, Kyu Nam Kim, Bong Soo Lee.

**Writing – original draft:** Soo Yeon Kim, Kyu Nam Kim, Mi Ae Jeong.

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
