## [Decision Letter · Decision Letter 0]

8 Feb 2021

PONE-D-20-39862

Ultrasound-guided Dynamic Needle Tip Positioning Technique for Radial Artery Cannulation in Elderly Patients: A Prospective Randomized Controlled Study

PLOS ONE

Dear Dr. Kim,

Thank you for submitting your manuscript to PLOS ONE. After careful consideration, we feel that it has merit but does not fully meet PLOS ONE’s publication criteria as it currently stands. Therefore, we invite you to submit a revised version of the manuscript that addresses the points raised during the review process.

Please carefully check the methodological comments and all other issues raise during revision.

We look forward to receiving your revised manuscript.

Kind regards,

Salvatore De Rosa

Academic Editor

PLOS ONE

Journal Requirements:

Reviewers' comments:

Reviewer's Responses to Questions

**Comments to the Author**

1. Is the manuscript technically sound, and do the data support the conclusions?

Reviewer #1: Yes

Reviewer #2: Yes

2. Has the statistical analysis been performed appropriately and rigorously? 

Reviewer #1: Yes

Reviewer #2: Yes

3. Have the authors made all data underlying the findings in their manuscript fully available?

Reviewer #1: Yes

Reviewer #2: Yes

4. Is the manuscript presented in an intelligible fashion and written in standard English?

Reviewer #1: Yes

Reviewer #2: Yes

5. Review Comments to the Author

Reviewer #1: A prospective randomized clinical trial was conducted to compare the ultrasound-guided DNTP technique with the palpation method in elderly patients. The first-attempt success rate and overall success rate in the DNTP group was significantly higher than the palpation group. Furthermore, the number of attempts and redirections were significantly lower in the DNTP group compared to the palpation group.

Minor revisions:

1- Line 180: For improved clarity, replace “number of samples” with “sample size.”

2- Line 186: Indicate the statistical testing method which archives 90% power.

3- Line 193: If data is nonparametric, express summary values as medians, first and third quartiles rather than means and standard deviations.

4- Table 4: Layout the columns to clearly distinguish between DNTP, palpation, and total hematoma.

Reviewer #2: In this trial the authors aim compare the ultrasound-guided Dynamic Needle Tip Positioning (DNTP) technique with the palpation method in elderly patients (>65 yo). The authors should be congratulated for this interesting trial. The manuscript is well written and technically sound.

After a careful evaluation of the paper these are my concerns:

1. Outcomes, sample size, sequence generation, allocation concealment, blinding, outcomes and estimation, harms, registration, and protocol are properly stated, while source of founding is not clearly stated. Conversely, the IRB approval is not in English, so this reviewer is not able to confirm this task.

2. The authors should specify why the authors have used the cutoff of 65 years to define elderly population, as 75 years is the mostly used selection criteria for this population

3. The authors should emphasize the novelty of this manuscript since data evaluating this technique have been already published.

4. The authors stated that “Although several studies have compared the use of ultrasound with the conventional palpation method for radial artery cannulation, (9-17) none of them investigated the ultrasound-guided DNTP technique in elderly patients. We hypothesized that the DNTP technique would have a higher success rate and reduce the incidence of complications”. However, comparing DNTP versus palpation is not appropriate to prove that DNTP is more effective than the available ultrasound techniques. Please address.

5. Please specify how many patients have been excluded.

6. Please implement table 2 legend

7. In the sample size the authors expected a 50% rate for first-attempt success in the control group. However, the observed ret in the control group is 72%. Please explain such a difference.

6. PLOS authors have the option to publish the peer review history of their article (what does this mean?). If published, this will include your full peer review and any attached files.

Reviewer #1: No

Reviewer #2: No

---

## [Author Response · Author response to Decision Letter 0]

11 Mar 2021

5. Review Comments to the Author

Reviewer #1: 

A prospective randomized clinical trial was conducted to compare the ultrasound-guided DNTP technique with the palpation method in elderly patients. The first-attempt success rate and overall success rate in the DNTP group was significantly higher than the palpation group. Furthermore, the number of attempts and redirections were significantly lower in the DNTP group compared to the palpation group.

Minor revisions:

1- Line 180: For improved clarity, replace “number of samples” with “sample size.”

Reply: I appreciate your careful review. As you pointed out, “number of samples” corrected to “sample size.”

The comments were described as follows and the text which was changed was highlighted in BOLD 

(page 11, lines 185–186) 

The sample size was calculated by referring to a randomized controlled trial[10] comparing ultrasound-guided DNTP and the palpation method for radial artery cannulation

2- Line 186: Indicate the statistical testing method which archives 90% power.

Reply: Estimating the number of samples at significance level (α) 0.05 and power 90% (1-β) to detect a statistically significant difference, a total of 253 subjects was required.

The comments were described as follows and the text which was changed was highlighted in BOLD 

(page 11, lines 190–191) 

The number of patients needed to achieve an α error of 5% and a β error of 10% was 253.

3- Line 193: If data is nonparametric, express summary values as medians, first and third quartiles rather than means and standard deviations.

Reply: Thank you for your suggestion. We expressed all data as mean and standard deviations according the central limit theorem in the first submitted manuscript. I totally agreed with your suggestion and no parametric data should be expressed as medians, first and third quartiles. 

The comments were described as follows and the text which was changed was highlighted in BOLD 

(page 2, lines 38–40) 

Continuous data were compared using Student’s t-test or the Mann-Whitney U-test and expressed as mean (standard deviation) or median (first, third quartiles), respectively. 

(page 13, lines 213 Table 1) 

(page 14, lines 229 Table 2) 

4- Table 4: Layout the columns to clearly distinguish between DNTP, palpation, and total hematoma.

Reply: As you pointed out, I corrected the table.

(page 15, lines 247 Table 4)

 

Reviewer #2: 

In this trial the authors aim compare the ultrasound-guided Dynamic Needle Tip Positioning (DNTP) technique with the palpation method in elderly patients (>65 yo). The authors should be congratulated for this interesting trial. The manuscript is well written and technically sound.

After a careful evaluation of the paper these are my concerns:

1. Outcomes, sample size, sequence generation, allocation concealment, blinding, outcomes and estimation, harms, registration, and protocol are properly stated, while source of founding is not clearly stated. Conversely, the IRB approval is not in English, so this reviewer is not able to confirm this task.

Reply: I appreciate your careful review. There was no external fund in this study. This information was mentioned within cover letter (journal assistant will change the online submission form). As you pointed out, the original IRB study protocol was written in Korean. Actually, I attached the translated study protocol in first submission. I attach the translated IRB study protocol and IRB approval in English at this time. 

(page 22, lines 364-365)

Supporting information

S3 Text. Ethical committee approval.

S4 Original protocol.

2. The authors should specify why the authors have used the cutoff of 65 years to define elderly population, as 75 years is the mostly used selection criteria for this population

Reply: As you pointed out, we defined patients who were 65 or older as elderly population. The reason we made this decision was that many other medical studies set the elderly population to be 65 years or older. [32-34] I definitely agreed with your opinion. If we set the elderly population to be over 75 years old, it is thought that the participants enrolled in the study had more age-related arterial wall changes and tortuous arteries due to various underlying diseases. This is a factor that can influence the results. This is a limitation of our study and this has been described in limitation section. 

The comments were described as follows and the text which was changed was highlighted in BOLD 

(page 19, lines 332–335) 

Third, we defined patients who were 65 or older as elderly following many other medical studies.[32-34] If we set the elderly population to be 75 years or older, age-related arterial wall changes and tortuous arteries would have been more prevalent in the study population.

32. Hoffman GJ, Liu H, Alexander NB, Tinetti M, Braun TM, Min LC. Posthospital Fall Injuries and 30-Day Readmissions in Adults 65 Years and Older. JAMA network open. 2019;2(5):e194276. 

33. Guralnik JM, Eisenstaedt RS, Ferrucci L, Klein HG, Woodman RC. Prevalence of anemia in persons 65 years and older in the United States: evidence for a high rate of unexplained anemia. Blood. 2004;104(8):2263-8. 

34. Krumholz HM, Chen YT, Vaccarino V, Wang Y, Radford MJ, Bradford WD, et al. Correlates and impact on outcomes of worsening renal function in patients > or =65 years of age with heart failure. The American journal of cardiology. 2000;85(9):1110-3. 

3. The authors should emphasize the novelty of this manuscript since data evaluating this technique have been already published.

Reply: I appreciate your suggestion. Because elderly patients show decreased response to beta-receptor stimulation and increased systemic vascular resistance and sympathetic nervous system activity, they often show non-stable blood pressure and heart rate during anesthesia. However, arterial cannulation is often difficult in the hard-to-catheterize radial artery in elderly patients because they often have age-related arterial wall changes and tortuous arteries due to various underlying diseases. Although several studies to assess the use of ultrasound, none of them investigated the ultrasound-guided DNTP technique in elderly patients. Therefore, we hypothesized that the DNTP technique would have a higher success rate and reduce the incidence of complications when compared with the conventional palpation method in elderly patients. As a result, this prospective, randomized controlled trial firstly revealed the efficacy and safety of the ultrasound-guided DNTP technique in elderly patients. Ultrasound-guided radial artery cannulation with the DNTP technique significantly improved the first-attempt and overall success rates and complications compared to the conventional palpation method in elderly patients. We expect that use of ultrasound with DNTP increases the efficiency of radial artery cannulation and minimizes tissue damage and complications by reducing the number of times the needle passes through in elderly patients.

With your help, I would emphasize the novelty of this manuscript. 

The comments were described as follows and the text which was changed was highlighted in BOLD 

(page 5, lines 57–63) 

As elderly patients show decreased response to beta-receptor stimulation and increased systemic vascular resistance and sympathetic nervous system activity, they often show non-stable blood pressure and heart rate during anesthesia.[3] However, arterial cannulation is often difficult in the hard-to-catheterize radial artery in elderly patients because they often have age-related arterial wall changes and tortuous arteries due to various underlying diseases.[4-6] 

(page 5, lines 74–78) 

Although several studies have compared the use of ultrasound with the conventional palpation method for radial artery cannulation,[10-18] none of them investigated the efficacy and safety of the ultrasound-guided DNTP technique in elderly patients. We hypothesized that the DNTP technique would have a higher success rate and lower incidence of complications compared with the conventional palpation method. 

(page 19, lines 338–340) 

Despite these limitations, this prospective, randomized controlled trial revealed for the first time the efficacy and safety of the ultrasound-guided DNTP technique in elderly patients. 

(page 20, lines 342–347) 

Ultrasound-guided radial artery cannulation with the DNTP technique significantly improved the first-attempt and overall success rates and reduced the numbers of attempts, redirections, cannulation time, and complications in elderly patients compared to the conventional palpation method. We expect that use of ultrasound with DNTP will increase the efficiency of radial artery cannulation and minimize tissue damage and complications by reducing the number of needle passes in elderly patients.

4. The authors stated that “Although several studies have compared the use of ultrasound with the conventional palpation method for radial artery cannulation, (9-17) none of them investigated the ultrasound-guided DNTP technique in elderly patients. We hypothesized that the DNTP technique would have a higher success rate and reduce the incidence of complications”. However, comparing DNTP versus palpation is not appropriate to prove that DNTP is more effective than the available ultrasound techniques. Please address.

Reply: I really appreciate your suggestion. I definitely agree with your opinion. This study compared DNTP versus conventional palpation method. 

The comments were described as follows and the text which was changed was highlighted in BOLD 

(page 5, lines 76–78) 

We hypothesized that the DNTP technique would have a higher success rate and lower incidence of complications compared with the conventional palpation method.

5. Please specify how many patients have been excluded.

Reply: Among 270 patients who were assessed for eligibility, 14 were excluded. Three patients declined to participate and 11 patients did not meet inclusion criteria (Hemodynamically unstable; 2 patients, Skin abnormalities such as inflammation or hematoma at the cannulation site; 3 patients, Abnormal results on the modified Allen test; 3 patients, History of hand or wrist surgery; 3 patients.) I also specify how many patients have been excluded in Figure 1. 

The comments were described as follows and the text which was changed was highlighted in BOLD 

(page 2, lines 38–40) 

Among 270 patients who were assessed for eligibility, 14 were excluded. Three patients declined to participate, and 11 patients did not meet inclusion criteria.

(Figure 1)

6. Please implement table 2 legend

Reply: Thank you for your careful review. I implemented table 2 legend. 

(page 14, lines 232-233) 

DNTP, dynamic needle tip positioning. SBPend, systolic blood pressure at the end of the procedure; HRend, heart rate at the end of the procedure; bpm, beats per minute.

7. In the sample size the authors expected a 50% rate for first-attempt success in the control group. However, the observed ret in the control group is 72%. Please explain such a difference.

Reply: As you pointed out, the palpation method had a higher first-attempt success rate (72.3%) than our expect (50%). This was probably due to the difference in number of radial arterial cannulations performed by the operators before the start of the study. The operators in the previous study placed at least 10 radial arterial cannulations using the DNTP technique and the palpation method prior to participation. On the other hand, the cannulation operators in our study performed arterial cannulation in at least 40 cases with either method to become familiar with the techniques. The minimum number of cases required was determined by previously reported learning curves and recommendations. According these studies, the learning curves revealed a marked improvement of arterial cannulation skill after 20 attempts, with a success rate ranging from 80%. The number of radial arterial cannulations was suggested as 40 attempts by IRB committee to ensure the patients safety. If we decided number of radial arterial cannulations as 10 attempts, we carefully predict that similar first-attempt success rate would be achieved. This is for the patient safety, please excuse this and we ask for your understanding. 

The comments were described as follows and the text which was changed was highlighted in BOLD 

(page 17, lines 295–303) 

The palpation method had a higher first-attempt success rate (72.3%) than expected based on a previous study (48%).[10] This was probably due to the difference in number of radial arterial cannulations performed by the practitioners before the start of the study. The practitioners in the previous study[10] placed at least 10 radial arterial cannulations using the DNTP technique and the palpation method prior to participation. The cannulation practitioners in our study performed arterial cannulation in at least 40 cases with either method to become familiar with the techniques. The minimum number of cases required was determined by previously reported learning curves and recommendations for ensuring patient safety.[19-22]

---

## [Decision Letter · Decision Letter 1]

12 Apr 2021

PONE-D-20-39862R1

Ultrasound-guided Dynamic Needle Tip Positioning Technique for Radial Artery Cannulation in Elderly Patients: A Prospective Randomized Controlled Study

PLOS ONE

Dear Dr. Kim,

Thank you for submitting your manuscript to PLOS ONE. After careful consideration, we feel that it has merit but does not fully meet PLOS ONE’s publication criteria as it currently stands. Therefore, we invite you to submit a revised version of the manuscript that addresses the points raised during the review process.

In particular, some improvement is still needed in statistical methods. Please, read carefully all comments and address them in a revised manuscript.

We look forward to receiving your revised manuscript.

Kind regards,

Salvatore De Rosa

Academic Editor

PLOS ONE

Journal Requirements:

Reviewers' comments:

Reviewer's Responses to Questions

**Comments to the Author**

1. If the authors have adequately addressed your comments raised in a previous round of review and you feel that this manuscript is now acceptable for publication, you may indicate that here to bypass the “Comments to the Author” section, enter your conflict of interest statement in the “Confidential to Editor” section, and submit your "Accept" recommendation.

Reviewer #1: (No Response)

Reviewer #2: All comments have been addressed

2. Is the manuscript technically sound, and do the data support the conclusions?

Reviewer #1: Yes

Reviewer #2: Yes

3. Has the statistical analysis been performed appropriately and rigorously? 

Reviewer #1: Yes

Reviewer #2: Yes

4. Have the authors made all data underlying the findings in their manuscript fully available?

Reviewer #1: Yes

Reviewer #2: Yes

5. Is the manuscript presented in an intelligible fashion and written in standard English?

Reviewer #1: Yes

Reviewer #2: Yes

6. Review Comments to the Author

Reviewer #1: Indicate the statistical testing method which archives 90% power.

Perhaps the method is the chi-square test for comparing proportions.

Reviewer #2: The authors have fully addressed my comments. The papers has been improved, well written and technically sound.

7. PLOS authors have the option to publish the peer review history of their article (what does this mean?). If published, this will include your full peer review and any attached files.

Reviewer #1: No

Reviewer #2: No

---

## [Author Response · Author response to Decision Letter 1]

13 Apr 2021

PONE-D-20-39862R1

Ultrasound-guided Dynamic Needle Tip Positioning Technique for Radial Artery Cannulation in Elderly Patients: A Prospective Randomized Controlled Study

PLOS ONE

Dear Dr. Kim,

Thank you for submitting your manuscript to PLOS ONE. After careful consideration, we feel that it has merit but does not fully meet PLOS ONE’s publication criteria as it currently stands. Therefore, we invite you to submit a revised version of the manuscript that addresses the points raised during the review process.

In particular, some improvement is still needed in statistical methods. Please, read carefully all comments and address them in a revised manuscript.

We look forward to receiving your revised manuscript.

Kind regards,

Salvatore De Rosa

Academic Editor

PLOS ONE

Journal Requirements:

Reply: We double-checked the references one by one. As a result, reference No. 8, and reference No. 22 have been described according to the journal references style. Because reference No. 11 could not be searched in Pubmed, we changed it with other appropriate references.  

Reviewers' comments:

Reviewer's Responses to Questions

Comments to the Author

1. If the authors have adequately addressed your comments raised in a previous round of review and you feel that this manuscript is now acceptable for publication, you may indicate that here to bypass the “Comments to the Author” section, enter your conflict of interest statement in the “Confidential to Editor” section, and submit your "Accept" recommendation.

Reviewer #1: (No Response)

Reviewer #2: All comments have been addressed

2. Is the manuscript technically sound, and do the data support the conclusions?

Reviewer #1: Yes

Reviewer #2: Yes

3. Has the statistical analysis been performed appropriately and rigorously?

Reviewer #1: Yes

Reviewer #2: Yes

4. Have the authors made all data underlying the findings in their manuscript fully available?

Reviewer #1: Yes

Reviewer #2: Yes

5. Is the manuscript presented in an intelligible fashion and written in standard English?

Reviewer #1: Yes

Reviewer #2: Yes

6. Review Comments to the Author

Reviewer #1: Indicate the statistical testing method which archives 90% power.

Perhaps the method is the chi-square test for comparing proportions.

Reply: I apologize for giving the wrong answer because I didn’t understand the previous question correctly. As you pointed out, χ2 test with a power of 90% and a significance level (α) of 0.05 was used. With your help, I was able to accurately describe the article. I appreciate your help. 

The comments were described as follows and the text which was changed was highlighted in BOLD. 

(page 11, lines 189–190) 

To detect a difference in the success rate using a χ2 test with a power of 90% and a significance level (α) of 0.05, a total of 253 subjects was required.

Reviewer #2: The authors have fully addressed my comments. The papers has been improved, well written and technically sound.

7. PLOS authors have the option to publish the peer review history of their article (what does this mean?). If published, this will include your full peer review and any attached files.

Do you want your identity to be public for this peer review? For information about this choice, including consent withdrawal, please see our Privacy Policy.

Reviewer #1: No

Reviewer #2: No

---

## [Editor Report · Decision Letter 2]

3 May 2021

Ultrasound-guided Dynamic Needle Tip Positioning Technique for Radial Artery Cannulation in Elderly Patients: A Prospective Randomized Controlled Study

PONE-D-20-39862R2

Dear Dr. Kim,

We’re pleased to inform you that your manuscript has been judged scientifically suitable for publication and will be formally accepted for publication once it meets all outstanding technical requirements.

Kind regards,

Salvatore De Rosa

Academic Editor

PLOS ONE
---

## [Editor Report · Acceptance letter]

6 May 2021

PONE-D-20-39862R2 

Ultrasound-guided Dynamic Needle Tip Positioning Technique for Radial Artery Cannulation in Elderly Patients: A Prospective Randomized Controlled Study 

Dear Dr. Kim:

I'm pleased to inform you that your manuscript has been deemed suitable for publication in PLOS ONE. Congratulations! Your manuscript is now with our production department. 

Kind regards, 

on behalf of

Dr. Salvatore De Rosa 

Academic Editor

PLOS ONE